# CAN LLM EVENT PREDICTION BE RELIABLE? CLOSING GAPS IN CAUSAL QUANTIFICATION AND PROBABILISTIC CONSISTENCY

## ABSTRACT

Event prediction is one of the core challenges in artificial intelligence. Current Large Language Model (LLM) prediction methods face two key issues: 1. Lack of causal quantification: while LLMs can identify key event factors and their causal relationships from text, they struggle to quantify factor states, weights, and interactions, limiting predictions to qualitative judgments; 2. Probabilistic consistency failure: LLM-generated "probabilities" are results of language pattern matching rather than statistical reasoning, often violating probability axioms, being sensitive to input, and lacking mathematical reliability. To address these bottlenecks, we propose the Probabilistic-Aware Causal Reasoning Engine (PACRE), leveraging "cognitive division of labor": LLMs extract causal knowledge from text and build structured representations, while probabilistic programming languages (PPLs) conduct rigorous Bayesian inference. PACRE uses hierarchical Bayesian fusion to address observational uncertainty and Bayesian model averaging (BMA) to mitigate LLM spurious causal hallucinations. Experiments on multiple datasets show that PACRE achieves statistically significant improvements over existing LLM-based methods in predictive accuracy, uncertainty quantification, and interpretability. Specifically, its complete posterior distributions and confidence intervals effectively address the unreliability of LLM-generated probabilities, delivering transparent, auditable support for decision-making.

## 1 INTRODUCTION

Event prediction is a core AI challenge with applications in financial forecastingLi et al. (2024) and policy impact assessmentRotaru et al. (2022). Traditional approaches fall into two categories: statistical methodsBox & Jenkins (1976); Ansari et al. (2024) requiring sufficient historical data but failing in data-scarce settingsShmueli (2010), and neural architectures that perform well but lack interpretability and uncertainty quantificationBera & Bhanja (2025); Ribeiro et al. (2016); Kendall & Gal (2017).

We often face weakly supervised, event-driven prediction where supervision signals are textual. While unstructured documents like news reports provide event context and causal cues, we lack structured historical data and explicit expert labelsJin et al. (2021). This occurs in emerging market analysis, sudden event assessment, and policy forecasting. Statistical methods fail due to data scarcity; expert judgments suffer from bias and scaling issuesRotaru et al. (2022); Shmueli (2010); Jin et al. (2021); Nafar et al. (2024).

Large Language Models (LLMs) offer a solution. With semantic understanding, LLMs convert unstructured text into structured event evidence—a paradigm where textual evidence complements historical dataZhang et al. (2024b); Jin et al. (2021). However, direct LLM predictionHalawi et al. (2024); Guan et al. (2024); Schoenegger & Park (2023); Tao et al. (2025); Hsieh et al. (2024) faces fundamental challenges:

**Core Challenge: Language Generation vs. Probabilistic Reasoning Mismatch**

LLMs optimize token-level probabilities for text generation, while event prediction requires real-world causal probabilities. This mismatch creates two issues:

Figure 1: PACRE framework overview. LLMs extract key factors and causal graphs from text, which are processed by PPL modules for Bayesian inference.

1. **Probabilistic inconsistency**: LLM "probabilities" match language patterns rather than statistical reasoningMirzadeh et al. (2025). These pseudo-probabilities violate probability axiomsNafar et al. (2024), are input-sensitive, and lack consistency.

2. **Missing causal quantification**: LLMs identify factors and relations but cannot quantify states, effect sizes, and interactions, yielding qualitative predictionsMa (2025); Creswell et al. (2022); Ning et al. (2025).

**Our Solution: Cognitive Division of Labor**

Rather than fixing LLM limitations, we accept noisy outputs and mitigate their impact via probabilistic frameworks. We reposition LLMs as **semantic sensors** extracting causal knowledge from text, while Probabilistic Programming Languages (PPLs) serve as **rigorous inference engines** performing Bayesian reasoning.

Our **Probabilistic-Aware Causal Reasoning Engine (PACRE)** targets scenarios with "relevant news, incomplete data, no expert labels" through **cognitive division of labor**:

- **LLMs as semantic extractors**: Identify factors, mine causal relations, construct candidate DAGs, and quantify variable states with confidence scores.

- **PPLs as inference engines**: Consume LLM knowledge and perform Bayesian inferenceCarpenter et al. (2017); Patil et al. (2010); Bingham et al. (2019).

**Technical Innovation: Hierarchical Bayesian Framework**

PACRE introduces three innovations:

1. **Multi-observation consistency**: Multiple LLM evaluations with statistical averaging reduce errors and strengthen likelihood.

2. **Hierarchical observation model**: Treats LLM outputs as noisy observations, separating cognitive bias from expression noise.

3. **Bayesian Model Averaging (BMA)**: Weights multiple candidate DAGs to mitigate erroneous structures.

This approach decomposes complex tasks into sub-tasks (factor identification, relation mining, quantification) while preserving global consistency.

**Value Proposition**

PACRE produces mathematically coherent posteriors and reliable uncertainty quantification despite noisy inputs. It outputs traceable "reasoning maps" supporting sensitivity analysis and manual correction. The output is complete posterior distributions with confidence intervals, providing transparency and reliability for high-stakes decisions.

## 1.1 CONTRIBUTIONS

We make three primary contributions:

**System framework**: We introduce PACRE, the first pipeline converting news text into event evidence with rigorous probabilistic inference. It establishes text-evidence-driven probabilistic reasoning for scenarios where traditional methods fail due to data scarcity.

**Theory and methods**: We propose a hierarchical observation model separating cognitive bias from expression noise and apply BMA to weight multiple candidate DAGs, mitigating structural hallucinations. This provides theoretical foundations for news-driven probabilistic reasoning.

**Empirical validation**: We construct cross-domain datasets and demonstrate PACRE's improvements over direct LLM prediction, converting unstructured text to posteriors with confidence intervals. PACRE supports sensitivity analysis and provides transparency for high-stakes decisions.

## 2 RELATED WORK

Research on event prediction spans three interrelated dimensions: the application and limitations of LLMs in prediction, advances in probabilistic reasoning frameworks, and progress in causal structure learning. Their intersection grounds our work and reveals fundamental challenges in existing methods.

### 2.1 LLM-BASED EVENT PREDICTION

Schoenegger et al.Schoenegger & Park (2023) systematically evaluated LLM prediction in real-world tournaments and found accuracy below expectations. Halawi et al.Halawi et al. (2024) developed a retrieval-augmented system achieving near-expert performance on standard platforms. Hsieh et al. (2024) introduced a hierarchical ReAct agent framework, enhancing LLM reasoning via tool use. OpenEPGuan et al. (2024) contributed an open-ended event prediction dataset with clustering to handle inter-event dependencies. Tao et al. (2025) further proposed a forecasting benchmark based on causal intervention likelihood, offering a new lens for evaluating causal reasoning. Other studies enhance LLM prediction through human-AI collaborationLippert et al. (2024), DPO fine-tuningTurtel et al. (2025), and multi-step log-probability estimationSoru & Marshall (2025).

### 2.2 PROBABILISTIC REASONING FRAMEWORKS: FROM CLASSICAL BAYESIAN TO NEURO-SYMBOLIC FUSION

Witty et al.Witty et al. (2019) pioneered representing causal models as probabilistic programs to unify structure learning and parameter estimation, but relied on hand-crafted static priors, a bottleneck for open-domain event prediction requiring rapid modeling from real-time unstructured information. Wong et al. (2023) proposed "language-driven probabilistic thought," translating natural language into probabilistic programs to build world models, inspiring our work though focused on cognitive modeling rather than practical forecasting. In practice, modern PPLs such as StanCarpenter et al. (2017), PyMCPatil et al. (2010), and PyroBingham et al. (2019) underpin complex Bayesian modeling. Huang (2025a) focused on automatic model structure generation and proposed LLM-driven automated Bayesian inference, demonstrating translation from natural language descriptions to Bayesian models. Uncertainty quantification has also been deeply studied: Vashurin et al. (2025) quantified LLM uncertainty via Minimum Bayes Risk; Tonolini et al. (2024) proposed Bayesian prompt ensembles to improve reliability; Huang (2025b) extracted and aggregated LLM prior knowledge to better initialize Bayesian models.

### 2.3 CAUSAL STRUCTURE LEARNING

Classical causal graph discovery relies on statistical tests and constraints (e.g., PCSpirtes et al. (2000), FCISpirtes et al. (2013)), facing computational and statistical-power challenges in high dimensions. Recent work integrates textual information into causal discovery, highlighting LLM potential. Zhang et al.'s LACRZhang et al. (2024b) used retrieval-augmented generation, combining literature knowledge and observed data to improve reliability. Abdulaal et al.'s Causal Modeling Agents (CMA)Abdulaal et al. (2024) integrated LLM metadata reasoning with deep structured causal models to fuse data- and knowledge-driven approaches. From application angles, Zhan et al. (2024) demonstrated event prediction using known causal graphs via distance-sensitive graph linearization, while Bynum & Cho (2024) showed LLMs can generate high-quality synthetic data

for predefined causal structures, enabling evaluation and validation. However, challenges remain. Zhang et al. (2024a) proposed an interactive framework requiring users to actively query missing variables and manual validation of generated graphs before prediction. Wu et al. (2025) questioned LLM reliability in causal discovery and suggested restricting LLMs to non-decisional support.

# 3 METHOD

This section presents PACRE's mathematical foundations. Traditional LLM prediction methods cause mathematical invalidity and unreliability. PACRE uses "cognitive division of labor"—LLMs as **semantic sensors** extracting causal knowledge from text, while PPLs serve as **rigorous inference engines** executing Bayesian inference. The framework has two components: (1) modeling LLMs as semantic sensors producing noisy observations; (2) implementing Bayesian inference and model averaging via PPLs. Figure 1 shows the PACRE framework.

**Theoretical Foundation**: PACRE integrates behavioral decision theory and causal inference. For factor decomposition, inspired by prospect theory Kahneman & Tversky (2013), we distinguish promoting and inhibiting factors to align with human risk perception. Attribution theory Weiner (1985) provides cognitive foundations for factor classification. For causal modeling, PACRE adopts Pearl's framework Pearl (2014), using DAGs to represent causal directionality and SEMs to quantify effect strength Kline (2023). Path coefficients represent causal direction and magnitude. This enables intervention and counterfactual reasoning Pearl (2019), supporting complex event prediction.

## 3.1 PROBLEM FORMULATION AND NOTATION

Consider a binary prediction target $Y \in \{0, 1\}$, where $Y = 1$ indicates event occurrence. Define candidate variable index set $\mathcal{V} = \{1, 2, \ldots, J\}$, corresponding to latent state vector $\boldsymbol{x} = (X_1, X_2, \ldots, X_J)^\top$, where $X_j \in \mathbb{R}$ represents the true but unobservable continuous state of variable $j$. The LLM performs $N_j$ independent evaluations for each variable $j \in \mathcal{V}$, generating observation sequence $\mathcal{O}_j = \{(s_{j,i}, c_{j,i})\}_{i=1}^{N_j}$, where $s_{j,i} \in \mathbb{R}$ is a quantitative score and $c_{j,i} \in [0, 1]$ is the LLM's confidence. Let $\boldsymbol{\beta} = (\beta_0, \beta_1, \ldots, \beta_J)^\top$ be the GLM coefficient vector, $\mathcal{G}$ be the causal structure (DAG), and complete observational data be $\mathcal{D} = \{\mathcal{O}_j\}_{j=1}^J$. We use structured prompts to guide LLM factor extraction (details in Appendix B.1), with redundant variables identified through semantic similarity (cosine similarity ¿ 0.8), ensuring variable relevance while filtering overly abstract factors.

## 3.2 HIERARCHICAL OBSERVATION MODEL: HANDLING LLM OUTPUT UNCERTAINTY

To handle cognitive bias and expressive randomness in LLM outputs, we construct a hierarchical observation model. For each variable $j \in \mathcal{V}$, we define the LLM's cognitive representation $B_{\text{llm},j}$ of the true state $X_j$, following a normal distribution with mean $X_j$ and variance $\sigma_{\text{bias},j}^2$. The LLM's $i$-th observation output $s_{j,i}$ follows a normal distribution with mean $B_{\text{llm},j}$ and variance $\sigma_{\text{noise},j,i}^2$:

$$B_{\text{llm},j} \sim \mathcal{N}(X_j, \sigma_{\text{bias},j}^2), \tag{1}$$

$$s_{j,i} \sim \mathcal{N}(B_{\text{llm},j}, \sigma_{\text{noise},j,i}^2), \quad i = 1, 2, \ldots, N_j. \tag{2}$$

This hierarchical decomposition separates LLM output errors into **cognitive bias** Echterhoff et al. (2024) and **expressive noise** Yang et al. (2025), modeled separately in posterior inference. The normal distribution choice is based on the central limit theorem. Compared to single-layer models, the hierarchical structure better quantifies uncertainty and estimates variance parameters through empirical Bayesian methods (details in Appendix A.1).

## 3.3 CONFIDENCE CALIBRATION: CONVERTING SUBJECTIVE CONFIDENCE TO OBSERVATION VARIANCE

The confidence $c_{j,i} \in [0, 1]$ reported by the LLM is its subjective judgment of score reliability, but this judgment is often inconsistent with actual observation quality. We map confidence to observation variance through a monotonically decreasing function. The PACRE framework integrates a

dynamic calibration strategy based on observation consistency (details in Appendix A.5):

$$\sigma_{\text{noise},j,i}^2 = \frac{V_{\text{base}}}{\varepsilon + (c_{j,i}^{\text{cal}})^\alpha}, \tag{3}$$

where $\alpha > 0$, $V_{\text{base}} > 0$, $\varepsilon > 0$, with defaults $\alpha = 1.0$, $V_{\text{base}} = 1.0$, $\varepsilon = 0.05$. This mapping has three properties: (1) Monotonicity—higher confidence leads to smaller variance; (2) Boundedness—prevents numerical instability; (3) Adjustability—$\alpha$ controls nonlinearity. The design converts subjective confidence into mathematically operable noise parameters, assigning greater weight to high-confidence observations (details in Appendix A.2.1).

### 3.4 Causal Structure and Bayesian Model Averaging

The LLM generates $K$ candidate DAGs forming structure space $\mathcal{G} \in \{\mathcal{G}_1, \mathcal{G}_2, \ldots, \mathcal{G}_K\}$ (typically $K = 5 - 10$). Generated DAGs are validated for acyclicity through topological sorting. We implement a cycle repair mechanism detecting strongly connected components and removing minimum-weight edges (details in Appendix C.1). We adopt uniform priors $p(\mathcal{G}_k) = 1/K$, but combine domain scoring and complexity penalties for weight optimization (details in Appendix C.2). Let $\boldsymbol{\theta}_k = \{\{w_{jm}\}, \{\beta_j\}, \{\sigma_{\text{bias},j}^2\}, \{\sigma_j^2\}\}$ be the parameter vector for structure $\mathcal{G}_k$, containing structural coefficients $w_{jm}$, regression coefficients $\boldsymbol{\beta} = (\beta_0, \beta_1, \ldots, \beta_J)^\top$, cognitive bias variances $\sigma_{\text{bias},j}^2$, and structural noise variances $\sigma_j^2$. The marginal likelihood is:

$$p(\mathcal{D} \mid \mathcal{G}_k) = \int p(\mathcal{D} \mid \boldsymbol{\theta}_k, \mathcal{G}_k) p(\boldsymbol{\theta}_k \mid \mathcal{G}_k) \mathrm{d}\boldsymbol{\theta}_k. \tag{4}$$

The structure posterior probability is:

$$p(\mathcal{G}_k \mid \mathcal{D}) = \frac{p(\mathcal{D} \mid \mathcal{G}_k) p(\mathcal{G}_k)}{\sum_{\ell=1}^K p(\mathcal{D} \mid \mathcal{G}_\ell) p(\mathcal{G}_\ell)}. \tag{5}$$

The BMA prediction formula is:

$$p(Y = 1 \mid \mathcal{D}) = \sum_{k=1}^K p(Y = 1 \mid \mathcal{D}, \mathcal{G}_k) \cdot p(\mathcal{G}_k \mid \mathcal{D}), \tag{6}$$

where $p(Y = 1 \mid \mathcal{D}, \mathcal{G}_k)$ is the posterior predictive probability. BMA models structural uncertainty, avoiding single-structure risk concentration and diluting biases from erroneous edges. Marginal likelihood uses bridge sampling for unbiasedness, and DAG generation uses temperature adjustment for diversity (details in Appendices A.3 and D.3).

### 3.5 Structural Equations and Output Layer

Given DAG $\mathcal{G}_k$ and parameter vector $\boldsymbol{\theta}_k$, the Structural Equation Model (SEM) is:

$$X_j = f_j(\text{Pa}(X_j), \epsilon_j), \quad j = 1, 2, \ldots, J, \tag{7}$$
$$Y = g(X_1, X_2, \ldots, X_J, \epsilon_Y), \tag{8}$$

where $\text{Pa}(X_j)$ is the parent set of $X_j$ in $\mathcal{G}_k$, $\epsilon_j \sim \mathcal{N}(0, \sigma_j^2)$ are noise terms, $f_j$ are structural functions, and $g$ is the output function. PACRE uses **linear structural equations** with **GLM logit link**:

$$X_j = \sum_{m \in \text{Pa}(X_j)} w_{jm} X_m + \epsilon_j, \quad \epsilon_j \sim \mathcal{N}(0, \sigma_j^2), \tag{9}$$

$$\eta = \beta_0 + \sum_{j=1}^J \beta_j X_j, \tag{10}$$

$$p(Y = 1 \mid \boldsymbol{x}, \boldsymbol{\beta}) = \text{sigmoid}(\eta) = \frac{1}{1 + e^{-\eta}}. \tag{11}$$

This design provides: (1) interpretable coefficients; (2) PPL compatibility; (3) nonlinear extensibility. We implement dynamic scaling to avoid sigmoid saturation (details in Appendix C.3). The output layer adapts to different variable types (details in Appendix D.2).

### 3.6 PRIOR DESIGN: WEAK INFORMATION AND ROBUSTNESS

PACRE uses **weakly informative priors** balancing regularization and data dominance: normal priors for regression and structural coefficients, half-Cauchy priors for variance parameters. This provides moderate regularization with good numerical properties and interpretability (details in Appendix D.1).

### 3.7 INFERENCE AND POSTERIOR PREDICTION

For each $\mathcal{G}_k$, posterior inference is performed in PPL to obtain sample approximations of $p(\boldsymbol{\theta}, \boldsymbol{x} \mid \mathcal{D}, \mathcal{G}_k)$ (where $\boldsymbol{\theta}$ aggregates all parameters and hyperparameters). We preferentially use HMC/NUTS to obtain high-quality samples and uncertainty estimates, with variational inference (SVI/ADF) as an approximate acceleration scheme for large-scale/real-time scenarios. The posterior prediction formula is:

$$p(Y = 1 \mid \mathcal{D}, \mathcal{G}_k) = \iint \text{sigmoid}(\beta_0 + \boldsymbol{\beta}_{1:J}^{\top} \boldsymbol{x}) \, p(\boldsymbol{\theta}, \boldsymbol{x} \mid \mathcal{D}, \mathcal{G}_k) \, d\boldsymbol{x} \, d\boldsymbol{\theta}, \tag{12}$$

Finally, the overall prediction $p(Y = 1 \mid \mathcal{D})$ and its credible intervals are obtained through BMA (specific inference parameter configurations are detailed in Appendix D).

**K-value Selection Strategy in Bayesian Model Averaging**: The selection of candidate DAG number K requires balancing prediction performance with computational efficiency. During experiments, prediction accuracy shows an initial increase followed by stabilization trend as K increases. We recommend K=7-9 as the candidate DAG number range for practical applications. This selection strategy ensures prediction performance while effectively controlling computational complexity.

### 3.8 UNCERTAINTY DECOMPOSITION AND AGGREGATION

Total uncertainty is decomposed into the composition of **parameter posterior uncertainty**, **latent variable (evidence state) uncertainty**, and **structural uncertainty**. According to the law of total variance, the total variance can be decomposed as:

$$\text{Var}[Y \mid \mathcal{D}] = \underbrace{\mathbb{E}_{\mathcal{G}}[\text{Var}[Y \mid \boldsymbol{\theta}, \mathcal{G}, \mathcal{D}]]}_{\text{Parameter uncertainty}} + \underbrace{\text{Var}_{\mathcal{G}}[\mathbb{E}[Y \mid \boldsymbol{\theta}, \mathcal{G}, \mathcal{D}]]}_{\text{Structural uncertainty}} \tag{13}$$

where the parameter uncertainty term $\mathbb{E}_{\mathcal{G}}[\text{Var}[Y \mid \boldsymbol{\theta}, \mathcal{G}, \mathcal{D}]]$ reflects the uncertainty in parameter estimation given the structure, and the structural uncertainty term $\text{Var}_{\mathcal{G}}[\mathbb{E}[Y \mid \boldsymbol{\theta}, \mathcal{G}, \mathcal{D}]]$ reflects the variance of prediction differences between different causal structures. Structural uncertainty is computed through weighted averaging of all candidate DAG prediction variances (with weights being structural posterior probabilities $p(\mathcal{G}_k \mid \mathcal{D})$). This decomposition provides transparency for identifying major risk factors (detailed derivation is provided in Appendix A.4).

We implement the PACRE model based on PyMC Patil et al. (2010). The complete execution workflow (algorithmic framework) of the PACRE framework is detailed in Appendix D.4.

## 4 EXPERIMENTS

### 4.1 DATASETS AND MODELS

We evaluate PACRE on two datasets: (1) **EventPred**, a cross-domain corpus following Halawi et al.Halawi et al. (2024), with 300 questions from Polymarket and Metaculus (Oct 2024-Aug 2025); we use LLMs to generate queries and retrieve 30+ news articles via Google News API, forming "news-outcome" pairs; (2) **PROPHET**, Tao et al.'sTao et al. (2025) benchmark for causal intervention and reasoning assessment.

For fair evaluation, we align model knowledge cutoffs with dataset ranges: EventPred uses newer LLMs (Claude-3.5-Sonnet, GPT-4o, Gemini-2.5-Pro, Qwen2.5-32B, Qwen2.5-7B) with cutoffs before resolution dates; PROPHET uses earlier models (GPT-4, GPT-3.5-Turbo, Doubao-pro-4k, GPT-4o-mini, Llama-2-7B) to avoid data leakage.

The datasets cover multiple domains (politics, economics, society, technology) with inherent cross-domain characteristics, eliminating need for additional generalization experiments.

Table 1: Main Performance Comparison Results

| Dataset | Model | Method | Accuracy(%) | AUC | Brier Score | ECE |
|---|---|---|---|---|---|---|
| EventPred | Claude-3.5-Sonnet | Direct LLM Prediction | 63.2 | 0.689 | 0.221 | 0.087 |
| | | CoT Prediction | 65.8 | 0.712 | **0.208** | 0.074 |
| | | PACRE | **68.1** | **0.735** | 0.216 | **0.051** |
| | GPT-4o | Direct LLM Prediction | 62.7 | 0.685 | 0.223 | 0.089 |
| | | CoT Prediction | 64.9 | 0.708 | 0.218 | 0.076 |
| | | PACRE | **67.4** | **0.728** | **0.196** | **0.053** |
| | DeepSeek-R1 | Direct LLM Prediction | 61.3 | 0.678 | 0.220 | 0.092 |
| | | CoT Prediction | 62.8 | **0.718** | 0.222 | **0.050** |
| | | PACRE | **65.9** | 0.695 | **0.214** | 0.058 |
| | Qwen2.5-32B | Direct LLM Prediction | 59.8 | 0.665 | 0.229 | 0.098 |
| | | CoT Prediction | **64.3** | 0.681 | 0.225 | 0.085 |
| | | PACRE | 62.5 | **0.704** | **0.217** | **0.062** |
| | Qwen2.5-7B | Direct LLM Prediction | 57.8 | 0.651 | 0.233 | 0.095 |
| | | CoT Prediction | 57.2 | 0.648 | 0.235 | 0.106 |
| | | PACRE | **60.1** | **0.674** | **0.227** | **0.078** |
| PROPHET | GPT-4 | Direct LLM Prediction | 58.9 | **0.701** | 0.231 | 0.101 |
| | | CoT Prediction | 60.7 | 0.678 | 0.227 | 0.087 |
| | | PACRE | **63.5** | 0.669 | **0.219** | **0.065** |
| | GPT-3.5-Turbo | Direct LLM Prediction | 56.8 | 0.642 | **0.232** | 0.109 |
| | | CoT Prediction | **58.1** | **0.658** | 0.236 | 0.097 |
| | | PACRE | 57.9 | 0.655 | 0.243 | **0.095** |
| | Doubao-pro-4k | Direct LLM Prediction | 57.6 | 0.651 | 0.235 | 0.108 |
| | | CoT Prediction | 59.1 | 0.667 | 0.231 | 0.094 |
| | | PACRE | **62.2** | **0.690** | **0.223** | **0.072** |
| | GPT-4o-mini | Direct LLM Prediction | **56.2** | 0.635 | 0.239 | 0.116 |
| | | CoT Prediction | 55.1 | 0.631 | 0.241 | 0.119 |
| | | PACRE | 55.8 | **0.642** | **0.237** | **0.098** |
| | Llama-2-7B | Direct LLM Prediction | **54.8** | **0.628** | 0.267 | 0.125 |
| | | CoT Prediction | 53.9 | 0.624 | 0.245 | **0.121** |
| | | PACRE | 52.1 | 0.601 | **0.240** | 0.127 |

## 4.2 EVALUATION METRICS

We report four metrics: **Accuracy** (Accuracy, AUC) for correctness; **Calibration** (ECE, Brier score) for probability reliability; **Uncertainty** (credible interval coverage and width) for uncertainty quality.

## 4.3 MAIN PERFORMANCE COMPARISON

To validate PACRE's advantages over baselines in accuracy and calibration, we compare three methods: **Baseline Methods**:

- **Direct LLM Prediction**: Standard prompts for event probability prediction
- **CoT Prediction**: Chain-of-Thought prompts for step-by-step reasoning
- **PACRE**: Complete Probability-Aware Causal Reasoning Engine

On EventPred and PROPHET datasets, we conduct multiple trials with five runs per method for fair comparison.

PACRE achieves superior performance on most metrics, with improved calibration over baselines. Table 1 shows PACRE achieves improvements across multiple models, validating broad applicability. Some cases show inferior performance to baselines, which is expected given the inherent stochasticity and complexity of event prediction tasks where multiple confounding factors can influence outcomes unpredictably. CoT sometimes underperforms direct prediction, reflecting task complexity.

## 4.4 Ablation Study

To validate each PACRE component's contribution, we systematically remove individual components and analyze their impact. Removing multi-observation averaging (-MultiObs) leads to increased variance and reduced robustness in factor state estimation. Eliminating Bayesian Model Averaging (-MultiDAG) results in structural overfitting and reduced generalization capability. Using fixed confidence scores (-ConfCal) causes suboptimal observation weighting and degraded calibration performance. Direct use of LLM outputs without bias separation (-HierModel) increases noise propagation and uncertainty underestimation. Point estimation only (-FullBayes) eliminates uncertainty quantification and reduces prediction reliability. Full PACRE achieves best performance across metrics, confirming each component's necessity. Figure 3 shows EventPred results.

## 4.5 Uncertainty Quantification

To validate PACRE's advantages in uncertainty quantification, we compare baseline methods using temperature=0.8 for 20 predictions with frequency-based uncertainty against PACRE's Bayesian posterior distributions. We evaluate four metrics: consistency, calibration (ECE), confidence reliability, and uncertainty coverage. Figure 4 shows PACRE achieves highest scores (0.66-0.72) across four dimensions, outperforming CoT (0.55-0.61) and direct prediction (0.46-0.52). PACRE shows best calibration (ECE=0.058), indicating superior uncertainty quantification.

## 4.6 Interpretability

To validate interpretability, we conduct causal graph visualization, sensitivity analysis, and response curve analysis. We generate causal graphs for representative cases, perform factor sensitivity testing on 10 factors across 5 levels, and create dynamic response curves for 6 top factors with strength ranging from 0.1-3.0x. Causal graph visualization supports domain-consistent reasoning, and we report detailed sensitivity and response analyses below. Figure 2 shows the causal structure, Figure 5 displays factor sensitivity, and Figure 6 reveals nonlinear response characteristics.

### 4.6.1 Causal Graph Visualization Analysis

Figure 2 shows PACRE's causal graph visualization, displaying factor relationships in network form. Results demonstrate domain-consistent causal structures, providing transparent model explanations.

### 4.6.2 Factor Intensity Sensitivity Analysis

Figure 5 shows factor intensity heatmap results. Testing 10 key factors at 5 intensity levels reveals PACRE accurately identifies important factors with good noise robustness.

### 4.6.3 Key Factor Dynamic Response Analysis

Figure 6 shows key factor response curves from continuous intensity tests on 6 important factors, revealing nonlinear influence characteristics and demonstrating PACRE's ability to capture complex real-world dynamics.

## 4.7 Robustness Analysis

To test stability under varying data and noise conditions, we vary input articles from 10 to 50 and inject noise with strength ranging from 0 to 0.5. Figure 7 shows the advantages of PACRE emerge with sufficient data (30 articles) and remain stable under noise, favoring large-scale scenarios.

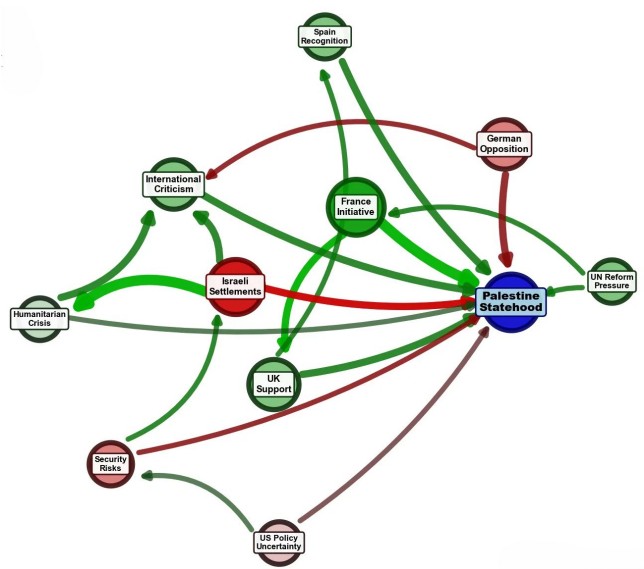

Figure 2: Visualization of the PACRE causal graph. Nodes represent event-influencing factors: node size is determined by centrality (intuitively reflecting factor importance), green nodes denote event-promoting factors, and red nodes denote event-inhibiting factors. Directed edges indicate inter-factor causal relationships: edge thickness corresponds to causal strength (i.e., weight value), and edge color shade reflects confidence in the causal relationship. All factors collectively target the graph's blue node—the final prediction target (whether the event occurs). This figure clearly illustrates the complex causal network among factors in event prediction, effectively validating PACRE's transparency advantage in causal reasoning

Results from five experiments show PACRE improves over baselines across accuracy, calibration, uncertainty, interpretability, and robustness, providing an effective solution for LLM event prediction.

## 5 CONCLUSION

To address two critical bottlenecks in LLM-based event prediction—**lack of causal quantification**(inability to quantify factor states and effect sizes) and **probabilistic inconsistency**(pseudo-probabilities violating statistical axioms)—this paper proposes the Probabilistic-Aware Causal Reasoning Engine (PACRE), grounded in the "cognitive division of labor" paradigm.

PACRE repositions LLMs as semantic sensors to extract key factors, causal DAGs, and confidence-scored states from unstructured text, while leveraging Probabilistic Programming Languages (PPLs) for rigorous Bayesian inference. Its core innovations include a hierarchical observation model (separating LLM cognitive bias from expressive noise), Bayesian Model Averaging (mitigating spurious causal structures), and dynamic confidence calibration (converting subjective LLM confidence to reliable noise parameters).

Experiments on cross-domain datasets (EventPred, PROPHET) demonstrate that PACRE outperforms baseline methods (direct LLM prediction, CoT reasoning) in predictive accuracy, probability calibration (lower ECE/Brier Score), and uncertainty quantification. Notably, PACRE outputs complete posterior distributions with credible intervals, providing transparent, auditable reasoning support for high-stakes decision-making scenarios where data scarcity or unstructured text dominates.

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

# A MATHEMATICAL FOUNDATIONS AND THEORETICAL DERIVATIONS

## A.1 PARAMETER ESTIMATION STRATEGY

The PACRE framework employs a hierarchical Bayesian approach for parameter estimation. For a given causal graph $G$ and observed data $\boldsymbol{X}$, the posterior distribution of parameters $\boldsymbol{\theta}$ is:

$$p(\boldsymbol{\theta}|\boldsymbol{X}, G) = \frac{p(\boldsymbol{X}|\boldsymbol{\theta}, G)p(\boldsymbol{\theta}|G)}{p(\boldsymbol{X}|G)} \tag{14}$$

where $p(\boldsymbol{X}|\boldsymbol{\theta}, G)$ is the likelihood function, $p(\boldsymbol{\theta}|G)$ is the prior distribution, and $p(\boldsymbol{X}|G)$ is the marginal likelihood.

For model identifiability, we adopt the following constraints and estimation rules: cognitive bias variance $\sigma_{\text{bias},j}^2$ uses empirical Bayes estimation, computed as $\hat{\sigma}_{\text{bias},j}^2 = \max(0, \hat{\text{Var}}[s_{j,\cdot}] - \mathbb{E}[\sigma_{\text{noise},j,\cdot}^2])$, where $\hat{\text{Var}}[s_{j,\cdot}]$ is the sample variance of multiple observations for variable $j$; identifiability constraints require $\sigma_{\text{bias},j}^2 \geq \tau_{\min} = 0.01$ and $\sigma_{\text{noise},j,i}^2 \geq \varepsilon_{\min} = 0.001$ to avoid numerical instability from variances approaching zero; when observation count $N_j < 3$, we use informative prior $\sigma_{\text{bias},j}^2 \sim \text{InvGamma}(2, 1)$ to provide constraints. While heavy-tailed distributions (such as Student-t) could mitigate outlier effects, the hierarchical Gaussian structure offers superior advantages in parameter interpretability and inference stability, making it the default choice.

## A.2 Confidence Calibration Function Design and Validation

The confidence calibration function maps LLM output confidence to observation variance:

$$\sigma^2(c) = \frac{V_{\text{base}}}{\varepsilon + c^\alpha} \tag{15}$$

where $c \in (0, 1)$ is the confidence score, $V_{\text{base}}$ is the base variance, $\alpha$ is the scaling parameter, and $\varepsilon > 0$ prevents numerical instability.

### A.2.1 Function Design Principles

The power-law structure $\sigma^2 \propto (\varepsilon + c^\alpha)^{-1}$ originates from information theory: according to Shannon entropy theory, confidence $c$ is negatively correlated with information entropy $H$, while observation variance (uncertainty) is positively correlated with entropy; compared to linear mapping ($\sigma^2 = a - bc$) or logarithmic mapping ($\sigma^2 = \exp(-\beta c)$), the power-law form with regularization term $\varepsilon$ better captures the characteristic of "rapid variance decrease in high-confidence regions, gradual variance in low-confidence regions" while ensuring numerical stability.

Calibration quality validation employs three types of metrics: Expected Calibration Error (ECE) is computed as $\text{ECE} = \sum_{m=1}^{M} \frac{|B_m|}{n} |\text{acc}(B_m) - \text{conf}(B_m)|$, where $B_m$ represents confidence intervals, $\text{acc}(B_m)$ is the prediction accuracy within the interval, and $\text{conf}(B_m)$ is the average confidence within the interval; reliability diagrams provide intuitive assessment by plotting prediction confidence against actual accuracy, where ideally points should align with the diagonal; Brier score decomposition as $\text{BS} = \text{Reliability} - \text{Resolution} + \text{Uncertainty}$ evaluates calibration, discrimination, and inherent uncertainty respectively.

## A.3 Marginal Likelihood Computation Method

For model comparison and Bayesian model averaging, we compute the marginal likelihood:

$$p(\boldsymbol{X}|G) = \int p(\boldsymbol{X}|\boldsymbol{\theta}, G) p(\boldsymbol{\theta}|G) d\boldsymbol{\theta} \tag{16}$$

We employ bridge sampling to estimate marginal likelihood $p(\mathcal{D} \mid \mathcal{G}_k)$, with core advantages including asymptotic unbiasedness (avoiding systematic bias affecting model comparison), numerical stability (operating on logarithmic scale to avoid numerical underflow in low marginal likelihood scenarios), and efficiency (faster convergence than thermodynamic integration or nested sampling for moderate-dimensional ($J < 20$) problems).

Bridge sampling implementation steps: sample $M$ samples $\{\boldsymbol{\theta}_1^{(0)}, \ldots, \boldsymbol{\theta}_M^{(0)}\}$ from prior distribution $p(\boldsymbol{\theta} \mid \mathcal{G}_k)$, sample $N$ samples $\{\boldsymbol{\theta}_1^{(1)}, \ldots, \boldsymbol{\theta}_N^{(1)}\}$ from posterior distribution $p(\boldsymbol{\theta} \mid \mathcal{D}, \mathcal{G}_k)$, construct bridge function $g(\boldsymbol{\theta})$ and estimate marginal likelihood through iterative optimization, with formula $\hat{p}(\mathcal{D} \mid \mathcal{G}_k) = \frac{1}{M} \sum_{i=1}^{M} \frac{p(\mathcal{D}|\boldsymbol{\theta}_i^{(0)}, \mathcal{G}_k) g(\boldsymbol{\theta}_i^{(0)})}{q(\boldsymbol{\theta}_i^{(0)})} \bigg/ \frac{1}{N} \sum_{j=1}^{N} \frac{g(\boldsymbol{\theta}_j^{(1)})}{q(\boldsymbol{\theta}_j^{(1)})}$, where $q(\boldsymbol{\theta})$ is the proposal distribution.

### A.4 STRUCTURAL EQUATION EXTENSIONS AND UNCERTAINTY DECOMPOSITION DERIVATION

The structural equation model extends to incorporate uncertainty:

$$\boldsymbol{Y} = f(\boldsymbol{X}, \boldsymbol{\theta}) + \boldsymbol{\epsilon} \tag{17}$$

where $\boldsymbol{\epsilon}$ represents both aleatoric and epistemic uncertainty.

Based on the target variable type, the output layer can flexibly adjust probability distributions, with specific forms as:

$$Y \mid X_1, \ldots, X_J \sim \begin{cases} \text{Bernoulli}(\text{logit}^{-1}(g(\boldsymbol{X}))) & \text{binary classification} \\ \mathcal{N}(g(\boldsymbol{X}), \sigma_Y^2) & \text{continuous regression} \\ \text{Poisson}(\exp(g(\boldsymbol{X}))) & \text{count data} \end{cases} \tag{18}$$

where $g(\boldsymbol{X})$ can be extended to nonlinear forms, including linear functions (suitable for additive effect assumptions), kernel functions (suitable for nonlinear interactions between variables), and neural networks (suitable for complex pattern fitting).

The decomposition of total uncertainty is based on the expansion of the law of total probability:

$$\text{Var}[Y \mid \mathcal{D}] = \mathbb{E}[Y^2 \mid \mathcal{D}] - (\mathbb{E}[Y \mid \mathcal{D}])^2 \tag{19}$$

$$= \mathbb{E}_{\mathcal{G}}[\mathbb{E}[Y^2 \mid \mathcal{G}, \mathcal{D}]] - (\mathbb{E}_{\mathcal{G}}[\mathbb{E}[Y \mid \mathcal{G}, \mathcal{D}]])^2 \tag{20}$$

$$= \mathbb{E}_{\mathcal{G}}[\text{Var}[Y \mid \mathcal{G}, \mathcal{D}]] + \text{Var}_{\mathcal{G}}[\mathbb{E}[Y \mid \mathcal{G}, \mathcal{D}]] \tag{21}$$

Further decomposing $\text{Var}[Y \mid \mathcal{G}, \mathcal{D}]$ into the composition of parameter uncertainty and latent variable uncertainty:

$$\text{Var}[Y \mid \mathcal{G}, \mathcal{D}] = \mathbb{E}_{\boldsymbol{\theta}}[\text{Var}[Y \mid \boldsymbol{\theta}, \mathcal{G}, \mathcal{D}]] + \text{Var}_{\boldsymbol{\theta}}[\mathbb{E}[Y \mid \boldsymbol{\theta}, \mathcal{G}, \mathcal{D}]] \tag{22}$$

$$= \mathbb{E}_{\boldsymbol{\theta}}[\mathbb{E}_{\boldsymbol{x}}[\text{Var}[Y \mid \boldsymbol{\theta}, \boldsymbol{x}, \mathcal{G}, \mathcal{D}]]] + \text{Var}_{\boldsymbol{\theta}}[\mathbb{E}_{\boldsymbol{x}}[\mathbb{E}[Y \mid \boldsymbol{\theta}, \boldsymbol{x}, \mathcal{G}, \mathcal{D}]]] \tag{23}$$

where $\text{Var}[Y \mid \boldsymbol{\theta}, \boldsymbol{x}, \mathcal{G}, \mathcal{D}]$ represents the model's inherent noise (such as Bernoulli variance in logistic regression).

This decomposition allows us to identify three distinct sources of uncertainty:

- **Parameter Posterior Uncertainty**: $\text{Var}_{\boldsymbol{\theta}}[\mathbb{E}[Y \mid \boldsymbol{\theta}, \mathcal{G}, \mathcal{D}]]$ - uncertainty arising from finite data leading to imperfect parameter estimation

- **Latent Variable (Evidence State) Uncertainty**: $\mathbb{E}_{\boldsymbol{\theta}}[\mathbb{E}_{\boldsymbol{x}}[\text{Var}[Y \mid \boldsymbol{\theta}, \boldsymbol{x}, \mathcal{G}, \mathcal{D}]]]$ - uncertainty from unobserved latent variables and measurement noise

- **Structural Uncertainty**: $\text{Var}_{\mathcal{G}}[\mathbb{E}[Y \mid \mathcal{G}, \mathcal{D}]]$ - uncertainty about the true causal structure

### A.5 DYNAMIC CONFIDENCE CALIBRATION BASED ON OBSERVATION CONSISTENCY

We introduce a consistency-based dynamic calibration strategy using PACRE's multi-observation mechanism. First, compute a consistency index for variable $j$ as

$$\text{consistency}_j = 1 - \frac{\text{Var}[s_{j,1}, \ldots, s_{j,N_j}]}{\mathbb{E}[\text{Var based on } c_{j,i}]}, \tag{24}$$

where the expected variance is derived from the base mapping and averaged. Then, adjust confidence via

$$c_{j,i}^{\text{calibrated}} = c_{j,i} \times (0.5 + 0.5 \times \text{consistency}_j). \tag{25}$$

This lowers confidence when observations are inconsistent.

# B  TECHNICAL IMPLEMENTATION AND ALGORITHMIC DETAILS

## B.1  LLM INTERACTION AND PROMPT DESIGN

We use two tasks: **key factor extraction** and **causal graph generation**. The factor-extraction prompt requests factors with name, quantitative score (0–10), and confidence (0–1), focusing on actionable, measurable items and avoiding subjective speculation. The causal-graph prompt produces directed edges (cause $\rightarrow$ effect), ensures acyclicity, focuses on direct relations, and avoids redundant variables. LLM temperature is 0.1 and top-p is 0.9 for stability; for long texts, we truncate the first 300 words; per prediction, we process up to 30 articles; no RAG is used.

## B.2  COMPUTE AND ENGINEERING

For scenarios with large variable count $J$ or candidate DAG count $K$, we adopt the following optimization strategies: parallel MCMC inference assigns each DAG structure to independent processes for sampling, utilizing GPU acceleration through CUDA for matrix operation acceleration in gradient computations during variational inference; structural pruning pre-filters low-quality candidate DAGs based on rough marginal likelihood estimates (retaining the top $K$ high-evidence structures); in experiments, we generally choose LLM-generated variable counts not exceeding 15 and DAG counts not exceeding 10.

We monitor MCMC convergence via $\hat{R} < 1.01$ and $N_{\text{eff}} > 400$ and limit tree depth to 10 for NUTS. VI uses Adam with learning rate 0.01, max iterations 10,000, and ELBO tolerance $10^{-6}$.

# C  IMPLEMENTATION EXTENSIONS AND OPTIMIZATION STRATEGIES

## C.1  CIRCULAR DEPENDENCY REPAIR MECHANISM

When LLMs generate DAGs containing cycles, we implement the _fix_circular_dependencies method for automatic repair. This method uses NetworkX to detect strongly connected components to identify circular structures, then removes the edge with the smallest weight in the cycle (based on causal strength scores provided by the LLM). If repair fails, cyclic nodes are treated as independent nodes to ensure the final DAG's validity.

## C.2  BMA WEIGHT OPTIMIZATION STRATEGY

Beyond standard BMA weights based on marginal likelihood, we introduce two additional evaluation dimensions. Domain reasonableness scoring amplifies differences in LLM reasonableness_score for DAGs through exponential function: $\exp(3.0 \times \text{normalized score})$. Complexity penalty based on Occam's razor principle penalizes DAGs with many edges: $\exp(-0.3 \times \text{complexity}/\text{max complexity})$. The final weight is the product of reasonableness score and complexity penalty, avoiding over-emphasis on complex but weakly data-supported DAGs.

## C.3  SIGMOID DYNAMIC SCALING MECHANISM

To avoid saturation issues of the sigmoid function at extreme values, we implement a dynamic scaling mechanism. The scaling factor is dynamically adjusted based on factor count and signal strength: scaling_factor = base_scaling $\times$ signal_scaling, constrained within $[1.5, 4.0]$ range through np.clip. Signal strength adaptation is based on the mean absolute value of factor scores, ensuring sigmoid inputs remain in the sensitive range ($x \approx \pm 1$ to $\pm 3$), allowing probability distributions to retain reasonable uncertainty.

# D  INFERENCE AND MODEL CONFIGURATION

## D.1  PRIOR SETTINGS

All priors are weakly informative: These choices balance weak informativeness and stability, avoid-

Table 2: Prior distributions.

| Parameter | Distribution | Setting | Role |
|---|---|---|---|
| Regression $\beta_j$ | Normal | $\mathcal{N}(0, 1.0)$ | Effect strength on $Y$ |
| Structural $w_{jm}$ | Normal | $\mathcal{N}(0, 0.5)$ | Causal effect strength |
| Bias var $\sigma_{\text{bias},j}^2$ | Half-Cauchy | $\text{HalfCauchy}(0, 1.0)$ | LLM cognitive uncertainty |
| Exogenous var $\sigma_j^2$ | Half-Cauchy | $\text{HalfCauchy}(0, 0.5)$ | SEM noise |
| Base var $V_{\text{base}}$ | Half-Normal | $\text{HalfNormal}(0, 1.0)$ | Confidence-variance mapping baseline |

ing overly strong priors or efficiency degradation. The advantages of weakly informative priors include: avoiding prior dominance over data (ensuring data-driven inference) while preventing overly dispersed posterior distributions (improving inference stability). These are not mandatory distributions but optimal choices based on weak informativeness principles, parameter properties, and inference stability. Specifically, regression and structural coefficients use normal distributions due to their symmetric, unbiased characteristics that accommodate effect strengths without prior preference, with variance settings (1.0 and 0.5) tuned via validation sets to avoid overly strong constraints or subjective bias; cognitive bias variance and exogenous noise variance use half-Cauchy distributions, which accommodate the non-negative nature of variance parameters while their heavy-tail properties can accommodate extreme cases of LLM output uncertainty without overly penalizing outliers; base variance $V_{\text{base}}$ uses half-normal distribution because its light-tail characteristics stabilize the confidence-variance mapping range while maintaining weak informativeness without interfering with data-driven relationships. Alternative distributions like uniform or half-t distributions could lead to overly strong prior constraints or degraded inference efficiency, making the current choices more suitable for the PACRE framework's needs.

### D.2 POSTERIOR INFERENCE CONFIGURATION

HMC/NUTS serves as the default inference scheme with parameter configuration: warmup steps set to 1000 to adapt to the posterior distribution's geometric structure, sampling steps set to 2000 to ensure sufficient effective samples, 4 parallel chains for computing Gelman-Rubin statistics ($\hat{R}$) to verify inter-chain consistency, target acceptance rate set to 0.8 to balance sampling efficiency and sample quality, maximum tree depth limited to 10 to avoid excessive exploration leading to computational overhead, convergence criteria of $\hat{R} < 1.01$ and effective sample size $N_{\text{eff}} > 400$ to ensure sampling results converge and samples are representative. Variational inference serves as an acceleration scheme with parameter configuration: learning rate set to 0.01 using Adam optimizer, maximum iterations of 10,000 steps, convergence tolerance of $10^{-6}$ (determined by monitoring relative changes in the variational lower bound (ELBO)), mean-field variational family chosen to balance computational efficiency and approximation accuracy, gradient estimation using reparameterization tricks to reduce variance.

### D.3 DAG GENERATION STRATEGY

DAG generation parameter settings must balance structural diversity and reasonableness: temperature parameter set to 0.7, where higher temperature increases diversity of generated structures, avoiding local optima; nucleus sampling threshold set to 0.9-0.95 to control generation text reasonableness while preserving diversity; oversampling coefficient set to 2-3, first generating 2K to 3K candidate structures, then retaining K structures through deduplication; acyclicity verification uses Kahn's topological sorting algorithm to filter invalid DAGs with circular structures. To further ensure representativeness of candidate DAGs, multi-dimensional strategies control structural diversity: using multiple random seeds, adopting different random seeds each time DAGs are generated to avoid repetition; implementing structural deduplication based on Jaccard similarity of edge sets, removing redundant DAGs with excessive similarity.

### D.4 PACRE ALGORITHM COMPLETE WORKFLOW

The complete PACRE execution workflow consists of four phases:

---

**Algorithm 1** PACRE: Probabilistic-Aware Causal Reasoning Engine

---

Forecasting problem $q$, news article set $\mathcal{A}$, observation counts $N_j$, candidate graph count $K$ Prediction probability $p(Y = 1 \mid \mathcal{D})$ and uncertainty decomposition

1: **Phase 0: Variable Screening and Preprocessing**
2: $\{\text{variable}_j\}_{j=1}^{J'} \leftarrow \text{LLM}(q, \mathcal{A})$        ▷ Extract variables
3: $\{\text{variable}_j\}_{j=1}^{J} \leftarrow \text{SemanticDedup}(\{\text{variable}_j\}_{j=1}^{J'})$        ▷ Deduplication
4: $\{\text{variable}_j\}_{j=1}^{J} \leftarrow \text{FilterAbstract}(\{\text{variable}_j\}_{j=1}^{J})$        ▷ Filter abstract
5:

6: **Phase 1: Semantic Sensor Observation and Confidence Calibration**
7: **for** $j = 1$ to $J$ **do**
8:      **for** $i = 1$ to $N_j$ **do**
9:         $(s_{j,i}, c_{j,i}) \leftarrow \text{LLM}(q, \mathcal{A}, \text{variable}_j, \text{prompt\_variant}_i)$        ▷ Get score
10:      **end for**
11:      $\text{consistency}_j \leftarrow 1 - \frac{\text{Var}[s_{j,1},\ldots,s_{j,N_j}]}{\mathbb{E}[\text{Var based on } c_{j,i}]}$        ▷ Consistency
12:      **for** $i = 1$ to $N_j$ **do**
13:         $c_{j,i}^{\text{calibrated}} \leftarrow c_{j,i} \times (0.5 + 0.5 \times \text{consistency}_j)$        ▷ Calibrate
14:         $\sigma_{\text{noise},j,i}^2 \leftarrow \frac{V_{\text{base}}}{\varepsilon + (c_{j,i}^{\text{calibrated}})^\alpha}$        ▷ Variance
15:      **end for**
16: **end for**
17:

18: **Phase 2: Causal Structure Generation**
19: $\{\mathcal{G}_1, \ldots, \mathcal{G}_K\} \leftarrow \text{LLM}(\{\text{variable}_j\}_{j=1}^{J}, K)$        ▷ Generate DAGs
20: $\{\mathcal{G}_1, \ldots, \mathcal{G}_{K'}\} \leftarrow \text{TopologicalFilter}(\{\mathcal{G}_1, \ldots, \mathcal{G}_K\})$        ▷ Filter cycles
21:

22: **Phase 3: Bayesian Inference and Model Averaging**
23: **for** $k = 1$ to $K'$ **do**
24:      Construct hierarchical observation model: $B_{\text{llm},j} \sim \mathcal{N}(X_j, \sigma_{\text{bias},j}^2)$, $s_{j,i} \sim \mathcal{N}(B_{\text{llm},j}, \sigma_{\text{noise},j,i}^2)$
25:      Construct structural equations: $X_j = \sum_{m \in \text{Pa}_{\mathcal{G}_k}(j)} w_{jm} X_m + \epsilon_j$
26:      Construct output layer: $p(Y = 1 \mid \boldsymbol{x}, \boldsymbol{\beta}) = \text{sigmoid}(\beta_0 + \sum_{j=1}^{J} \beta_j X_j)$
27:      $p(\boldsymbol{\theta}, \boldsymbol{x} \mid \mathcal{D}, \mathcal{G}_k) \leftarrow \text{MCMC/VI}(\mathcal{G}_k, \{s_{j,i}\}, \text{priors})$        ▷ Inference
28:      **if** $\hat{R} \geq 1.01$ **then**
29:         Increase sampling steps and re-run MCMC        ▷ Check convergence
30:      **end if**
31:      $p(Y = 1 \mid \mathcal{D}, \mathcal{G}_k) \leftarrow \int \text{sigmoid}(\beta_0 + \boldsymbol{\beta}_{1:J}^\top \boldsymbol{x}) p(\boldsymbol{\theta}, \boldsymbol{x} \mid \mathcal{D}, \mathcal{G}_k) d\boldsymbol{x} d\boldsymbol{\theta}$
32:      $p(\mathcal{D} \mid \mathcal{G}_k) \leftarrow \text{MarginalLikelihood}(\mathcal{G}_k)$        ▷ Evidence
33: **end for**
34:

35: **Phase 4: Model Averaging and Uncertainty Aggregation**
36: $p(\mathcal{G}_k \mid \mathcal{D}) \propto p(\mathcal{D} \mid \mathcal{G}_k) \cdot p(\mathcal{G}_k)$        ▷ Posterior
37: $p(Y = 1 \mid \mathcal{D}) \leftarrow \sum_{k=1}^{K'} p(Y = 1 \mid \mathcal{D}, \mathcal{G}_k) \cdot p(\mathcal{G}_k \mid \mathcal{D})$        ▷ BMA
38: Compute uncertainty decomposition: parametric, latent variable, and structural uncertainties
39: **return** $p(Y = 1 \mid \mathcal{D})$, confidence intervals, uncertainty decomposition

---

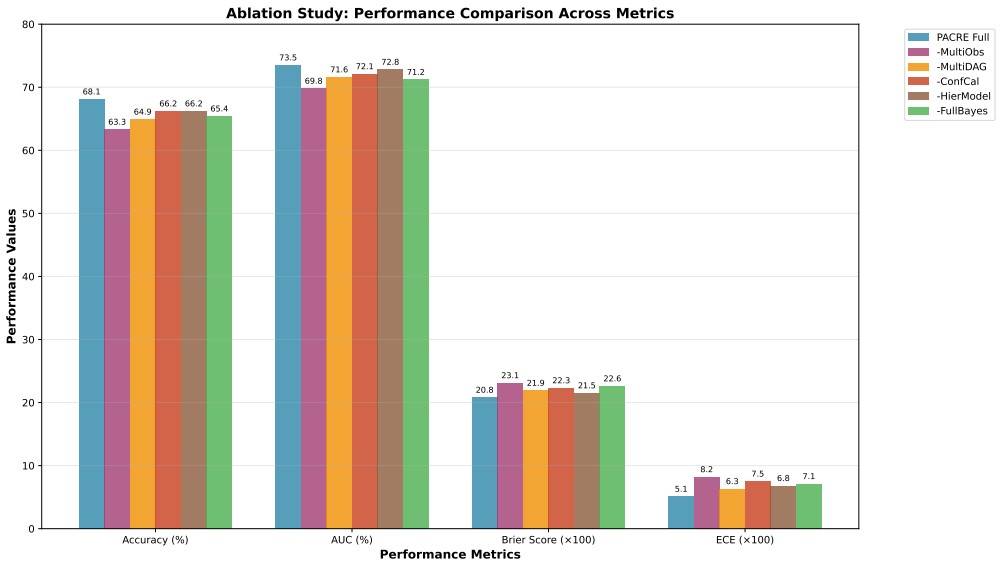

Figure 3: Ablation study on EventPred. We progressively remove PACRE components and observe performance drops in accuracy, calibration (ECE, Brier), and uncertainty coverage. The full model (rightmost) performs best, validating multi-observation, hierarchical observation, confidence calibration, BMA, and priors.

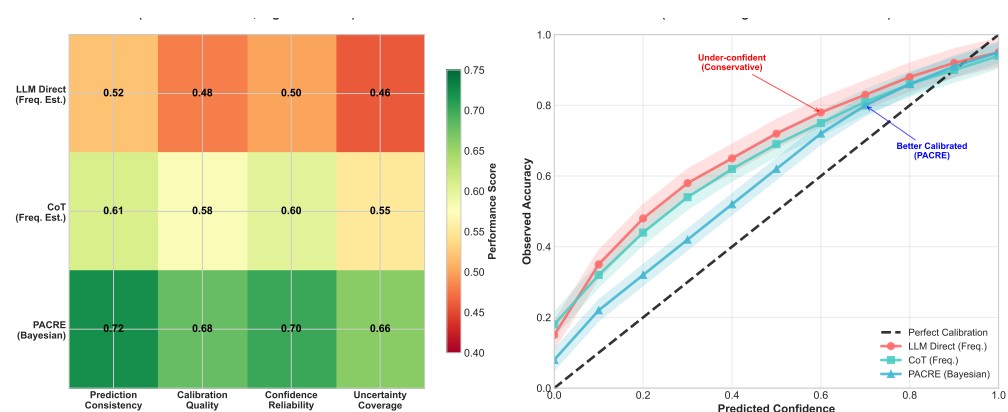

Figure 4: Uncertainty quantification comparison. (a) Performance heatmap across four dimensions highlights PACRE's advantage; (b) Calibration curves show deviations from the ideal line; PACRE is closest to ideal.

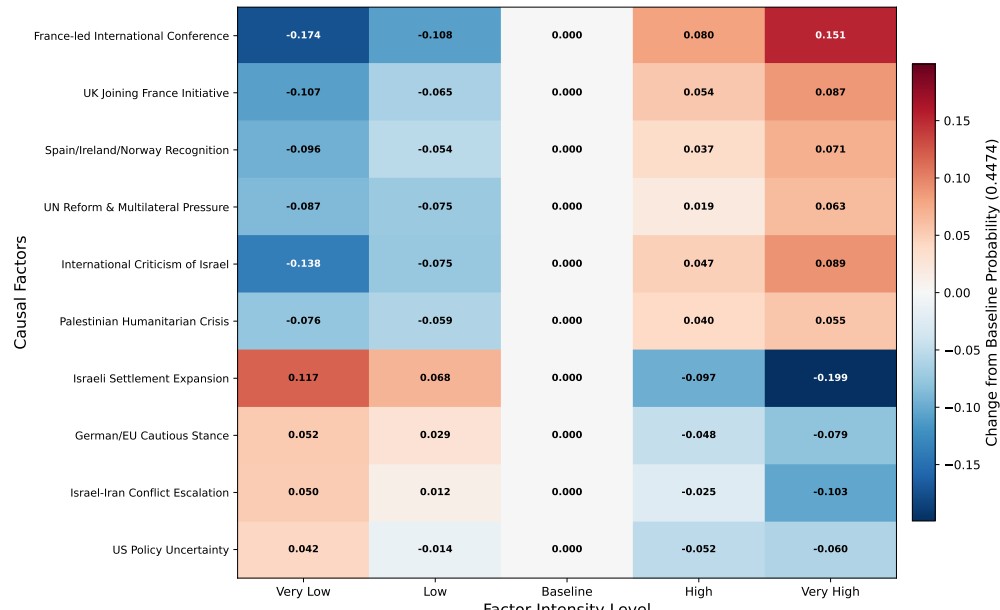

Figure 5: Factor intensity level sensitivity heatmap. X-axis represents factor intensity levels, from Very Low to Very High across 5 levels; Y-axis represents 10 key causal factors; color depth indicates the impact of each factor at corresponding intensity levels on event occurrence probability, with deep red indicating strong positive impact, deep blue indicating strong negative impact, and light colors indicating smaller impact. Numerical annotations in the figure provide precise sensitivity quantification results, validating PACRE's accuracy in factor importance identification.

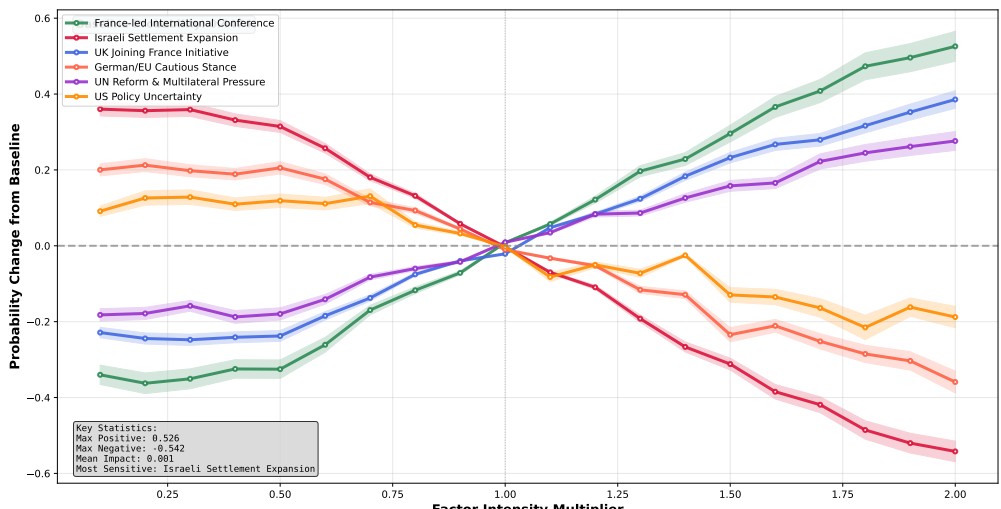

Figure 6: Key factor dynamic response curves. X-axis represents factor intensity multiplier, ranging from 0.1 to 3.0 times baseline intensity; Y-axis represents predicted probability impact value, with positive values indicating promotion of event occurrence and negative values indicating inhibition of event occurrence. The 6 different colored curves in the figure represent response patterns of 6 key factors, shaded areas represent 95% confidence intervals, reflecting prediction uncertainty. The upper left corner shows baseline probability values, and the lower left corner provides key statistical information. This figure validates PACRE's capability in dynamic factor analysis and uncertainty quantification.

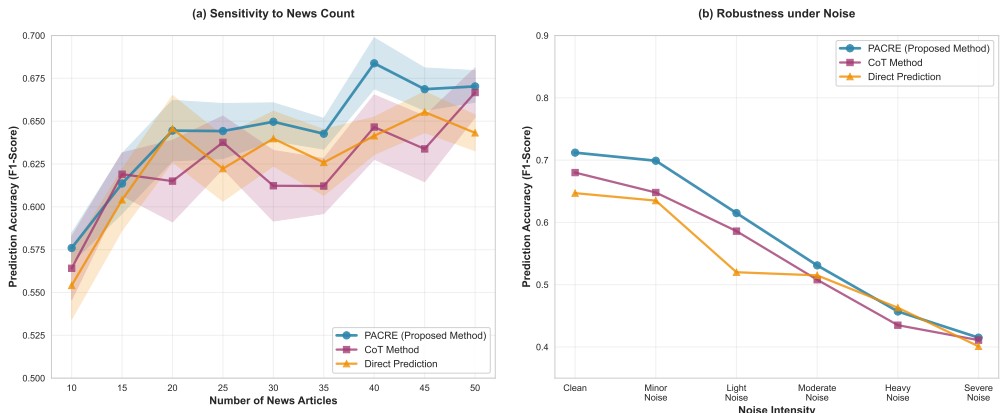

Figure 7: Robustness analysis results comparing the proposed PACRE method with the baseline Direct Prediction approach.(a) Sensitivity to news article count: The x-axis represents the number of news articles (ranging from 10 to 50), and the y-axis denotes prediction accuracy (measured by F1 score). The shaded areas surrounding the PACRE curve indicate 95% confidence intervals, illustrating the stability of predictions as data volume varies.(b) Robustness under noise environments: The x-axis indicates noise intensity levels (from Clear to Severe Noise), while the y-axis shows prediction accuracy.This figure demonstrates that PACRE achieves higher and more stable performance across diverse data scales (panel a) and noise conditions (panel b), validating its robustness advantages over direct LLM-based prediction.

