# OpenReview forum: "Can LLM Event Prediction Be Reliable? Closing Gaps in Causal Quantification and Probabilistic Consistency"
_ICLR.cc/2026/Conference — ICLR 2026 Conference Withdrawn Submission_

### Official Review · Reviewer_Adu6 · 2025-10-23

**Soundness:** 1
**Presentation:** 2
**Contribution:** 3
**Rating:** 2
**Confidence:** 5

**Summary:**

The paper explores how LLMs can support event prediction by converting unstructured text into structured event representations. The authors argue that LLMs possess semantic understanding useful for identifying causal factors and relationships from natural language but face challenges when their outputs are used for probabilistic reasoning.

The key challenges include:
* LLMs' statements of probabilities only reflect semantic plausibility rather than statistically valid quantities, often violating probability axioms.
* While LLMs can extract factors and relationships, they cannot quantify their strength, state, or interactions—limiting them to qualitative rather than quantitative event prediction.

The authors propose a framework called PACRE that combines:
* LLMs, acting as semantic sensors to extract entities, relations, and potential causal structures (candidate DAGs) from text, with
* Probabilistic modules to perform Bayesian reasoning over these extracted variables.

In general, PACRE models the process by which LLMs generate quantitative scores (confidence levels and event probabilities) as a linear model over latent state variables, then applies Bayesian inference to estimate these latent states and associated uncertainty.

**Strengths:**

The paper tackles a critical challenge in causal reasoning—the gap between semantic understanding and probabilistic inference. The motivation to align linguistic representations with causal probabilistic reasoning is well-grounded and timely for event forecasting tasks. The first half of the paper lays out the problem motivation clearly, thus justifying the need for a hybrid probabilistic-LLM approach.

**Weaknesses:**

Unfortunately, the second half of the paper is highly problematic in terms of both the methodology and the presentation. I summarize the key points here and detail them with questions in the next section.

**1. Ambiguous Problem Definition:**
* The notion of an “event” is not rigorously defined. Without clear semantics for what constitutes an event, it’s unclear how variables, state values, or factor relationships are instantiated, which makes the entire pipeline hard to evaluate or reproduce. The paper should provide a running example that demonstrates how variables of interests are defined from text data and a graphical method illustrating how the data-generative model is defined over these variables.

**2. Lack of Theoretical Rigor:**
* The proposed method is theoretically shallow. The probabilistic model connecting LLM-generated scores to latent states lacks formal justification. There is no discussion of identifiability or theoretical guarantees of the recovery of the true variables involved: the factors, their states and the other model parameters.

**3. Overly Simplistic Modelling Assumptions:**
* The assumed linear model for LLM score generation is too simplistic to capture complex dependencies in real-world event structures.
* Lack of discussion on the consequences under model/prior misspecification e.g., when assumptions about noise or latent structure are violated
* The method restricts the causal search space to graphs that LLMs can output, which may not even include the true causal structure (likely so since the DAG space is vast). This design choice undermines the claim of reliable uncertainty quantification and interpretability.

**4. Unclear Experimental Validation:**
* Evaluation settings are poorly described—it is unclear whether “accuracy” refers to numerical probabilities, textual event correctness, or causal structure quality. Due to this ambiguity, I cannot judge, from the presented empirical evidence, the effectiveness of the proposed method, in terms of the recovery of the true model and accuracy of the estimated quantities involved.

**5. Poor Presentation:**
* The paper seems to be in a draft state with several presentation issues e.g., citation spacing, font rendering error (line 193), undefined abbreviations or notations e.g., GLM in line 190, quantities in Eq. (3) like $V_{base}$, $c^{cal}$ etc.

**Questions:**

**1. Definition of Events:**
* How exactly is an “event” defined given text input?
* Are semantically equivalent sentences (e.g., “two people are talking” vs. “two people are speaking to each other”) treated as the same event or different ones?
* How are entity variations handled? For instance, is a “war between country A and country B” same as or different from a “war between country C and D”?

**2. Nature of State Variables:**
* What are the state variables in the latent model?
* Can the authors provide concrete examples showing how textual factors map to these states?
* How are redundant vs. relevant variables determined (Line 193 - 194)?

**3. Model Assumptions and Misspecification:**
* How sensitive is the model to misspecified likelihoods or nonlinear dependencies among variables?

**4. Evaluation:**
* What exactly does “accuracy” in Table 1 measure?
* Is the metric based on textual event matching or numeric probability estimation?

**5. Causal Graph Validation:**
* Has the accuracy of the output causal graph been assessed? I could not find such an experiment in the paper (already checked Appendix).
* If not done so, could the authors conduct experiments to validate the accuracy of the inferred causal structures? The authors could consider using CRAB dataset [1] with annotated causal relations of news data.


[1] Romanou, A., Montariol, S., Paul, D., Laugier, L., Aberer, K., & Bosselut, A. (2023, December). CRAB: Assessing the Strength of Causal Relationships Between Real-world Events. In Proceedings of the 2023 Conference on Empirical Methods in Natural Language Processing (pp. 15198-15216).

---

### Official Review · Reviewer_uCDB · 2025-10-27

**Soundness:** 2
**Presentation:** 2
**Contribution:** 2
**Rating:** 2
**Confidence:** 4

**Summary:**

The authors present PACRE which is a framework that combines LLMs and probabilistic programming languages (PPL) to improve event prediction. It uses LLMs to extract causal information and PPL to conduct Bayesian inference. Experiments on multiple datasets are shown to illustrate the efficacy of the proposed method.

**Strengths:**

1. Understanding LLMs and causal knowledge is an important research area and this work is welcome
2. The setup and goal of the framework is explained well

**Weaknesses:**

1. The organization of the paper could use some work, a bulk of the results is relegated to the appendix making it harder to understand PACRE's strengths and weaknesses
2. The size and scale of the datasets used is not large - raising scalability concerns
3. Captions of tables and figures could use a more detailed touch so that they are standalone

**Questions:**

1. Are the scores and confidence numbers from LLMs reliable?
2. Why are different models used for comparison between EventPred and PROPHET? This makes it harder to compare performance
3. What is the reasoning behind PACRE not performing well in some cases shown in Table 1?


Nit
1. You can use "``" to render opening quotes in LaTeX
2. Using OpenAI's proprietary logo in Figure 1 may attract unwanted attention. I'm not affiliated with them but do the authors have permission to do this?

---

### Official Review · Reviewer_FgLu · 2025-10-29

**Soundness:** 2
**Presentation:** 2
**Contribution:** 3
**Rating:** 4
**Confidence:** 4

**Summary:**

This paper addresses the failure of current LLMs to perform causal quantification and their generation of inconsistent probabilities. The authors propose the Probabilistic-Aware Causal Reasoning Engine (PACRE), a hybrid approach that uses LLMs to extract causal knowledge and PPLs to conduct  Bayesian inference. The method employs hierarchical Bayesian fusion to handle observational uncertainty and Bayesian model averaging  to mitigate LLM casual hallucinations. The authors report statistically significant improvements in both accuracy and uncertainty quantification.

**Strengths:**

The paper addresses a critical and under-studied problem: the difficulty LLMs have in generating accurate probabilities

A novel algorithm to address LLM’s limitation to perform bette casual reasoning

**Weaknesses:**

The justification for the design choice in 3.2 and 3.3 is very brief: 1) the justification for modeling the true state $X_j$ with a normal distribution based on the central limit theorem is not elaborated . 2) the "consistency-based dynamic calibration strategy" (Appendix A.5) appears to be an ad-hoc heuristic.

The paper lack clarity in terms of writing and presentation: 1) Key concepts like cognitive bias are mentioned without adequate introduction. HMC/NUTS is not cited. 2) The description of the evaluated datasets is unclear. 3) the main paper only presents a single experiment result, forcing readers to consult the appendix. 4) citation formatting error.

The evaluation lacks robustness with too few baselines: 1) the authors should at least report direct and CoT baseline with self-consistency. 2) the uncertainty baseline in Section 4.5 is weak, as it relies on a simple "frequency-based" method derived from only 20 predictions.

**Questions:**

1. What do the author mean by cognitive bias?
2. How do you inject noise with strength ranging from 0 to 0.5 in line 431?
3. How long it takes for the PACRE algorithm to run on instance?

---

### Official Review · Reviewer_8kje · 2025-11-01

**Soundness:** 1
**Presentation:** 1
**Contribution:** 3
**Rating:** 4
**Confidence:** 3

**Summary:**

In this paper, the authors propose a method that leverages LLMs to learn the causal structure within text and perform Bayesian inference, thereby enhancing the causal reasoning and consistency in event prediction tasks using LLMs. The authors conduct extensive experiments to validate the effectiveness of their proposed method.

**Strengths:**

1. Equipping LLMs with causal reasoning capabilities and statistical consistency is an important and valuable contribution to the field.

2. The proposed approach that first learns the causal structure and then performs Bayesian inference is well-motivated and logically sound, though there are some concerns regarding whether the specific implementation fully realizes this approach.

3. The ablation studies are extensive, effectively validating the necessity of each design choice.

**Weaknesses:**

1. The most significant weakness of this manuscript lies in its writing. While I appreciate the authors' clear description of the problem, the motivation behind the work, and the thorough experimental setup, the detailed implementation of the method and the theoretical foundations are too brief and underdeveloped. For instance: (1) In terms of implementation details, the paper lacks clarity on how the authors use LLM to generate candidate DAGs during causal structure learning. (2) Regarding the theoretical aspects, the Theoretical Foundation (lines 176-181) and the further explanations in the appendix are not rigorous formal proofs but rather sketches. This makes it difficult to assess the theoretical soundness of the proposed method. (3) There are also some minor issues, such as the improper use of \citet and \citep, the absence of citations of some used methods or software like HMC/NUTS and NetworkX, and the absence of spaces in several places, which affects the readability of the paper.

2. Perhaps due to the lack of clarity in presentation, the theoretical soundness of the proposed method remains questionable. (1) The proposed approach relies on LLMs for causal discovery. However, it is highly uncertain whether such an approach can accurately identify and estimate the true causal structure, particularly if causal discovery is performed solely based on textual semantics. In this case, LLMs can only serve as auxiliary tools to traditional causal discovery algorithms, as they do not inherently possess causal reasoning capabilities. (2) The authors claim that the mapping strategy in confidence calibration has three properties (lines 221-222), yet no rigorous mathematical proof is provided. The sketched proof in the appendix is insufficient to convincingly support these claims.

3. The proposed approach of first learning the causal structure and then performing Bayesian inference is very related to the idea presented in TabPFN [1], where TabPFN first generates causal structures and then performs Bayesian inference. Although the problems they aim to address are different, and TabPFN is more theoretically sound as it is a tabular model rather than a language model, it would be necessary for the authors to discuss the similarities and differences between the two approaches in their paper.

[1] Hollmann, N., Müller, S., Purucker, L., Krishnakumar, A., Körfer, M., Hoo, S. B., ... & Hutter, F. (2025). Accurate predictions on small data with a tabular foundation model. Nature, 637(8045), 319-326.

**Questions:**

Please see Weaknesses.

---

### Note · Authors · 2025-11-21

I have read and agree with the venue's withdrawal policy on behalf of myself and my co-authors.